# Mineralized belemnoid cephalic cartilage from the late Triassic Polzberg *Konservat-Lagerstätte* (Austria)

**Petra Lukeneder**[1]*, **Alexander Lukeneder**[2]

**1** University of Vienna, Doctoral School of Ecology and Evolution, Vienna, Austria, **2** Museum of Natural History Vienna, Vienna, Austria

* a0501032@unet.univie.ac.at

**Data Availability Statement:** All raw data and measurements are available in the Supporting Data File. 3D surface data will be made available upon publication in the https://www.pangaea.de/ data base (https://issues.pangaea.de/browse/PDI-

## Abstract

Although hyaline cartilage is widely distributed in various invertebrate groups such as sabellid polychaetes, molluscs (cephalopods, gastropods) and a chelicerate arthropod group (horseshoe crabs), the enigmatic relationship and distribution of cartilage in taxonomic groups remains to be explained. It can be interpreted as a convergent trait in animal evolution and thus does not seem to be a vertebrate invention. Due to the poor fossil record of cartilaginous structures, occurrences of mineralized fossil cartilages are important for evolutionary biology and paleontology. Although the biochemical composition of recent cephalopod cartilage differs from vertebrate cartilage, histologically the cartilages of these animal groups resemble one another remarkably. In this study we present fossil material from the late Triassic Polzberg *Konservat-Lagerstätte* near Lunz am See (Lower Austria, Northern Calcareous Alps). A rich Carnian fauna is preserved here, whereby a morphogroup (often associated with belemnoid remains) of black, amorphous appearing fossils still remained undetermined. These multi-elemental, symmetrical fossils show remarkable similarities to recent cartilage. We examined the conspicuous micro- and ultrastructure of these enigmatic fossils by thin-sectioning and Scanning Electron Microscopy (SEM). The geochemical composition analyzed by Microprobe and Energy Dispersive X-ray Spectroscopy (SEM-EDX) revealed carbonization as the taphonomic pathway for this fossil group. Mineralization of soft tissues permits the 3D preservation of otherwise degraded soft tissues such as cartilage. We examined eighty-one specimens from the Polzberg locality and seven specimens from Cave del Predil (formerly Raibl, Julian Alps, Italy). The study included morphological examinations of these multi-elemental fossils and a focus on noticeable structures like grooves and ridges. The detected grooves are interpreted to be muscular attachment areas, and the preserved branched system of canaliculi is comparable to a channel system that is also present in recent coleoid cartilage. The new findings on these long-known enigmatic structures strongly point to the preservation of cephalic cartilage belonging to the belemnoid *Phragmoteuthis bisinuata* and its homologization to the cephalic cartilage of modern coleoids.

31315) and on the website of the Polzberg Project (https://www.nhm-wien.ac.at/forschung/geologie__palaeontologie/forschungsprojekte/polzberg). Images or additional information are available upon request from Petra Lukeneder.

**Funding:** This work was created in the course of projects funded by the Austrian Academy of Sciences (ÖAW), represented by the National Committee for Geo/Hydro-Sciences (Earth Sciences Program), project Polzberg Lukeneder and the Federal Government of Lower Austria (Department Science and Research) project K3-F-964/001-2020. Project funding was acquired by AL. Open access funding for publication provided by University of Vienna (acquired by PL). Study design, field work and data collection was funded by the above projects. The authors are responsible for the contents of this publication. The funder had no impact on conceptualization, design, data collection, analysis, decision to publish, or preparation of the manuscript.

**Competing interests:** The authors have declared that no competing interests exist.

# Introduction

Hyaline cartilage as connective tissue with its supporting, skeletal function has a long evolutionary-biological history and was developed in many different animal lineages such as polychaetes, echinoderms and molluscs [1–5]. Within the molluscs, hyaline cartilage is reported as a radula-supporting structure in predatory gastropods such as *Busycon canaliculatum*[5] and various cephalopods [2–7]. The fossil record of cartilage and cartilaginous structures is sparse, but common for fossil chondrichthyans. First descriptions of cephalic cartilages are included early in cephalopod, paleontological research [8]. A review of fossil cephalopod cartilages is given by Donovan & Fuchs (2016) [37]. Cephalic cartilage of an Upper Jurassic octobrachian or vampyromorph cephalopod *Loliginites* (*Geoteuthis*) *zitteli* (= *Loligosepia aalensis*) reported from the Posidonia shale of Schömberg (Germany) [9–12]. Jurassic specimens from Solnhofen limestone comprise preserved cephalic cartilage including statocysts and other soft tissues [12–15]. The complex evolutionary relationships within the major cephalopod groups are topic of intense research [16–19]. The diagnostic characters for the Order Phragmoteuthida, include the three-lobed proostracum and morphology of arm armature, have been described blurry [20]. Their systematic position as sister taxon of the modern coleoids and thus stem group Octobrachia [17] is widely accepted [20].

From the late Triassic Polzberg *Konservat-Lagerstätte*, the preservation of belemnoid soft tissues and probable mandibles of *Phragmoteuthis* specimens, as well as of trachyceratid soft tissues, has already been addressed [21, 22]. This study examines historical fossil material and new findings from the Polzberg locality and Cave del Predil. Earlier research suggested that the enigmatic black structures from Cave del Predil were halves of belemnoid jaws [23, 24], belonging to *Phragmoteuthis bisinuata*. The presence of cartilaginous rings on the slabs was also proposed [23]. This theory was later adopted by numerous authors [Doguzhaeva, pers. com. 2012; 25, 26] but this contribution is the first to provide detailed evidence. In the present paper we describe the fossil structures from the Polzberg *Konservat-Lagerstätte* in detail and homologize them to structures in modern coleoids.

## The Polzberg *Konservat-Lagerstätte*

The paleontological site Polzberg has been known for almost 150 years and was reported synonymously under various names such as Unter Polzberg, Pölzberg, Polzberg-Graben, Schindelberggraben and Polzberg [26–28] and references therein. In early collections, material from deposits of the Reingraben Shales was not separated from the Lunz Formation, and analogous specimens from the Polzberg locality were often designated as Lunz locality [26, 27]. The fossil site is located 4.5 km north-east of Lunz am See, on the western slope of mount Schindelberg (1066 m), within the Reifling basin and belongs to the Bajuvaric Lunz Nappe System of the Northern Calcareous Alps (Fig 1). It is accessible from the north via Erlauftal street 25, then Zellerrain street 71 or from the south via Mariazell via the Zellerrain street 71. The exact GPS position is N 47˚53'5.90" and E 15˚ 4'27.70" (see also [26]; 712 m above sea level).

The IRIS system (Interaktives Rohstoffinformations System) hints to an early historical adit (No. 071/3008a) for the production of black coal, active in the first half of the 19th century, which was confirmed by the literature [28]. The outcrop Polzberg locality (Fig 2A) was first mentioned by [28]. Further historical adits for the recovery of fossil specimens were dug in 1885 (Geological Survey of Austria, GBA) and in 1909 (Natural History Museum Vienna, NHMW) by Joseph Haberfelner [25] and references therein at approx. GPS 47˚53'6.20"N, 15˚ 4'28.30"E (710 m a.s.l.; Fig 1). In early research, the deposits outcropping at Polzberg were assigned to the Wengen shales [28], bearing fossils such as "*Ammonites*" *aon*, the double-valved crustacean *Eusteria* sp. and specimens that belong to the highly variable taxon *Halobia minuta*.

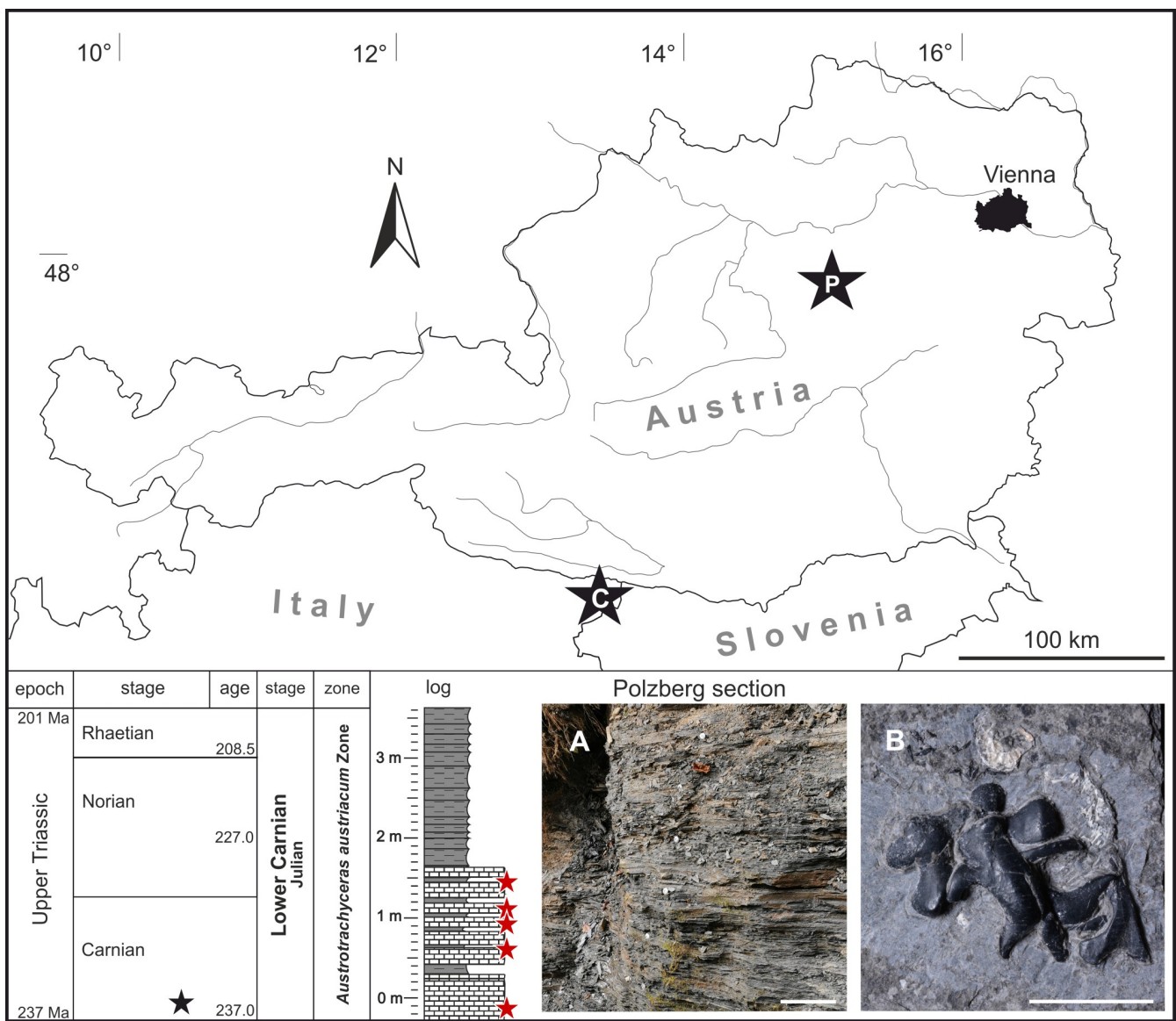

**Fig 1. Austrian map with stratigraphic and lithological overview and positions of layers (red asterisks) comprising the here described black carbonized fossils.** Black asterisk with P: position of the *Konservat-Lagerstätte* Polzberg (= Schindelberggraben ravine, Polzberg locality) near Lunz am See. Black asterisk with C: Cave del Predil, Italy comprising also deposits with enigmatic black fossils. **(A)** Vertical section (middle part) of Polzberg locality. Black asterisk marks position within the stratigraphic scale. Scale bar 20 cm. **(B)** Enigmatic carbonized structure from Polzberg locality, NHMW 2021/0001/0002. Scale bar 1 cm.

Belemnoid specimens of *Phragmoteuthis bisinuata*, fish remains of "*Belonorhynchus*" *striolatus* and plants assigned to *Voltzia haueri* were also reported [28]. Dark-grey to black, foliated clays and marls of the Reingraben Shales (= *Halobienschiefer*) intercalated by several limestone beds crop out at the Polzberg locality. Exceptionally preserved fossils are distributed within these sediments, pointing to a fully marine origin with sporadic influx of freshwater [29]. The older Reingraben Shales are thought to be a deep marine environment with nektonic (actively free-swimming) faunal elements [29]. For the Polzberg *Konservat-Lagerstätte*, dysoxic to anoxic conditions were proposed [27], while other authors favored a shallow marine, basinal environment [21]. The enigmatic fossils from Carnian deposits of Cave del Predil were described and figured in early works and assigned to the belemnoid *Phragmoteuthis bisinuata* [23]. Historical

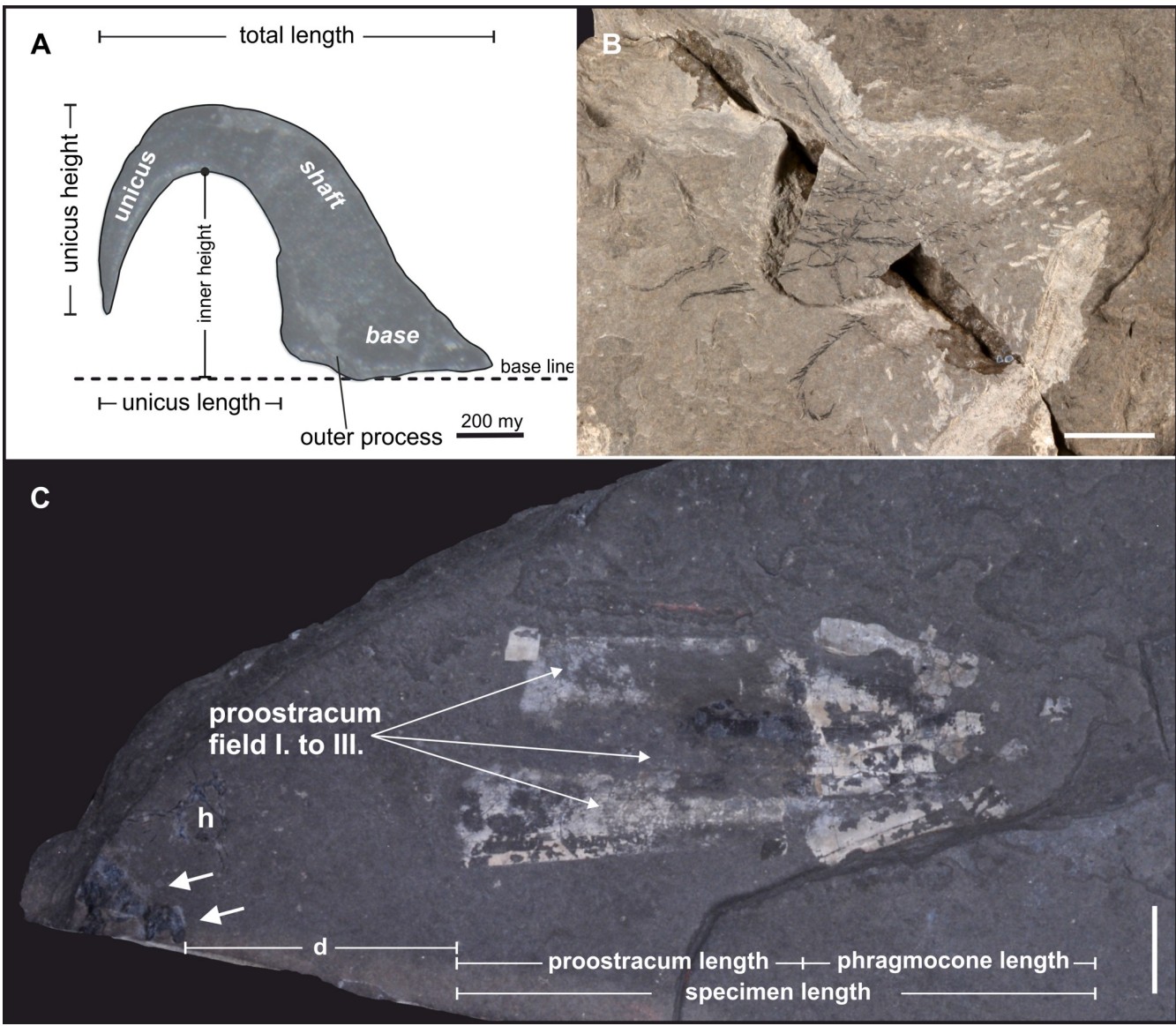

**Fig 2. Belemnoid fossils from the late Triassic *Konservat-Lagerstätte* Polzberg near Lunz am See. (A)** Hook morphology in phragmoteuthid arm crowns. **(B)** Semi in situ preserved arm crown. Each arm comprises two rows of hook armory. Terms adapted from [30]. **(C)** Belemnoid fossil remains, assigned to the taxon *Phragmoteuthis bisinuata* with its characteristic fan-shaped three-parted proostracum. White arrows pointing to position of carbonized structure.

excavations by the GBA and the NHMW, as well as recent findings by fossil collectors (2012–2014) and scientific field work in 2021, yielded similar specimens (Fig 1B) from Polzberg.

## The fossil record of cephalopod cephalic cartilage

Fossilization includes processes in which organic components can be replaced by inorganic minerals. Although the fossil record of cartilage ranges back to the Paleozoic [31], it is poor due to the rapid degradation of soft tissues. Cenozoic fossil cartilage, where the exceptional preservation state was linked to a surrounding phosphate-rich environment, can be examined using immunohistochemical and biochemical methods [32]. Calcified cartilage of the Campanian (Upper Cretaceous) *Hypacrosaurus stebingeri* was the subject of more recent

investigations [33, 34]. Early historical investigations assumed cephalic cartilage and its muscular elements in a specimen of *Loligosepia aalensis* [9] and described this in detail [10–12]. An Upper Carboniferous cephalic cartilage capsule was reported from Oklahoma [35]. Important findings on the *Dorateuthis syriaca* from *Konservat-Lagerstätten* in Lebanon enabled reconstructing the cephalic cartilage of this taxon [36]. A detailed compilation of fossilized cephalic cartilage findings within the genera was published [37]. Accordingly, indications for fossilized head cartilage were found in:

a. the phragmocone-bearing belemnoid genera *Acanthoteuthis* and possibly *Phragmoteuthis*,

b. the gladius-bearing Prototeuthina *Plesioteuthis* and *Dorateuthis*

c. the Teudopseina *Glyphiteuthis*, *Rachiteuthis* and *Muenstella*.

d. the extinct octopod genus *Keuppia*.

More recently, cephalic cartilage with statocyst remains in specimens of a Jurassic *Acanthoteuthis* from Solnhofen (Germany) were examined [14]. Fin cartilages of the middle Olenekian (Lower Triassic) *Idahoteuthis parisiana* from Idaho (USA) revealed a canalicular structure [38]. UV-light and light of visible wavelengths was useful in studying the cephalic cartilage of carboniferous cephalopods [38], where the orange color of parts of the head cartilage indicated phosphatized soft tissue.

## The evolution of invertebrate cartilage

Hyaline cartilage with its translucent appearance and the cartilage cells embedded in an extracellular, hydrophilic matrix occurs not only in vertebrates, but is also widely distributed in various lophotrochozoan groups [1] such as in some gastropod buccal masses, cephalopods, and sabellid polychaetes [1–3, 5, 6, 39–41]. The occurrence of real cartilage in horseshoe crabs is outstanding. The assumption is therefore that this tissue is not a vertebrate invention [3, 6, 41] but evolved convergently more than once. Although invertebrate cartilage differs biochemically from vertebrate cartilage[5], cephalopod and vertebrate cartilage share similar morphological features on the histological level [2, 5, 40, 42, 43]. Cephalopod cartilage even shares similarities to vertebrate bone [44]. The initial cartilaginous structures visible at the embryonic stage are the funnel cartilage and the cartilage of the pallial complex within the cephalopod brain [6]. Even in hatchlings, cephalic cartilage is visible as a thin layer of collagen in hatchlings of certain cephalopod groups such as *Loligo pealeii* [6]. In contrast to other recent cephalopods, here only the sepiid coleoids and the deep sea cephalopod group of *Spirula spirula* still have kept their mineralized phragmocone [45]. The predatory cephalopod mode of life requires an early development of additional supporting and protective functions of the ocular cartilages and an oculomotoric sensory system [6, 46]. Several cartilage types are associated with the eye [1]: cephalic cartilage, scleral cartilage and the equatorial ring ("iris-cartilage") [4, 40]. Ocular cartilage is present in several cephalopod groups such as *Nautilus*, *Octopus*, *Eledone*, *Sepia* and *Spirula* [5]. The scleral cartilage and the equatorial ring contain less matrix than the cephalic cartilage, which is also a less cellular type [4]. Conversely, eye-associated cartilage seems to be more cellular (containing less matrix) than cephalic cartilage [40].

## The structure and function of cephalic cartilage in modern coleoids

Thirteen cartilaginous structures are present in adult specimens of *Sepia officinalis*, most function as attachment points for muscular structures and are therefore important for the locomotory system [6]. Magnetic resonance imaging data of different cephalopods provides insights into the spectrum of cephalopod cephalic cartilage [47]. The supporting function of cranial

cartilage (= cephalic cartilage = head cartilage [1]) protects the squid's brain from external forces [4]. This structure is partially not closed [5], featuring orbital depressions [4, 5]. Extensive cephalic cartilage is present in several cephalopod groups such as *Octopus vulgaris*, *S. officinalis* (European cuttlefish) and *Loligo pealii* [4]. Optical ganglia of the nervous system and the eyes are housed in ocular cartilage, which can be interpreted as lateral parts of the cephalic cartilage [6]. These cartilaginous parts become fused in late developmental stages by bridging cartilage and statocysts, which contain the statoliths [6]. Extensive research [1, 3, 5, 6, 41, 42, 48, 49] shows that the cartilaginous ultrastructure of several cephalopod groups histologically resembles vertebrate cartilage. The funnel cartilage of *I. illecebrosus* strongly resembles vertebrate cartilage [49]. Cartilaginous structures such as the funnel cartilage, but also the cephalic cartilage in *S. officinalis*, are surrounded by a distinct, thin layer, the perichondrium [3]. The presence of cartilage-lining cells indicates cell growth from the central cartilage part [3]. The presence of a channel system is one histological feature which can be found in modern coleoid and vertebrate cartilage [6]. This system is often present in the extracellular matrix of cephalopod cephalic cartilage as a passage for blood vessels [1, 4]. Interestingly, the coleoid chondrocytes (cartilage cells) extend into long processes [6, 50, 51], a feature typical of vertebrate osteocytes but not for vertebrate chondrocytes. Furthermore, the morphology of cephalopod chondrocytes seems to change relative to their particular position within the cartilage [44].

Several authors reviewed the biochemical composition of recent cephalopod cartilage [4, 6]. The extracellular matrix of cephalopod cartilage consists of fibrous proteins in a hydrophilic ground substance [6]. The extracellular matrix of *Sepia* contains different types of collagen [40]. The staining of this matrix in *Illex illecebrosus* is unevenly distributed [40]

Various authors examined the mineralization processes of invertebrate cartilage in media metastable to hydroxylapatite at 37˚C [52] and references therein. The histology and mineralization of the cephalic cartilage of *L. pealii* was figured in [4]. A peripheral thickening in some squids was determined in reconstructions of the microanatomy and histology of the central nervous systems and of the eyes in coleoid hatchlings [53].

## Material and methods

### Material and institutional abbreviations

NHMW, Natural History Museum Vienna, GBA Geological Survey of Austria. No permits were required for the described study, which complied with all relevant regulations.

Digging permission from Franziska and Hermann Hofreiter (Gaming), the owner of the Polzberg section during the whole duration of the project. This study did not include live or anesthetized animals, cephalopods for morphological studies were obtained dead from fish market.

Eighty-six specimens (obtained from sixty-six samples) of the here described, carbonized fossils were available for analyses. A full list of examined specimens is provided in S4 Table. All specimens were drawn digitally and measured. One specimen was used for thin-sectioning (NHMW 2012/0117/0024), 13 specimens were analyzed by serial sectioning. 82 specimens stem from the Carnian Reingraben shales at the Polzberg locality (= Polzberggraben; historical and new findings). Bed-by-bed collected specimens found by the authors from distinct levels: Po -50–0 cm, Po 60–80 cm, Po 80–100 cm, Po 100–120 cm, Po 140–160 cm (Fig 2). Seven fossils come from Cave del Predil (northern Italy). The belemnoids from the Polzberg *Konservat-Lagerstätte* are compressed and mostly flattened, while the investigated black, enigmatic structures are not. The analyzed material is stored in the paleontological collection of the NHMW (81 specimens) and the GBA (five specimens). The fossil structures were analyzed with a variety of analytical tools as follows:

## Digital imaging and image processing

Macro-photography of all fossil specimens was done with a Nikon D 5200 SLR, lens Micro SX SWM MICRO 1:1 Ø52 Nikon AF-S, Digital Camera, in combination with the freeware graphic tool digiCamControl Version V.2.1.2.0 at the NHMW. High-resolution digital micro-photography was done using a Discovery.V20 Stereo Zeiss microscope, processed with the software AxioVision SE64 Rel. 4.9 imaging system at the NHMW.

## Scanning Electron Microscopy (SEM)

SEM images of the surface, as well as of the internal structures, of specimen NHMW 2012/0117/0024 were taken using the Quanta™ 250 FEG from FEI (with a Shottky field emission source FEG-ESEM) from the Department of Material Sciences and Process Engineering (MAP) at the University of Natural Resources and Life Sciences, Vienna. The electron microscope was equipped with an Everhardt Thornley SED-Detector, in low-vacuum settings with 15kV accelerating voltage. The specimen was therefore not gold-sputtered. Overview images were taken with a JEOL "Hyperprobe" JXA 8530-F field-emission electron microprobe (FE-EPMA) in combination with an online JEOL quantitative ZAF-correction program at the Central Research Laboratories of the NHMW. The sample was coated for EDS analyses with an 8 nm carbon film. An accelerating voltage of 15 and 5 keV, a beam current of 5 nA, and fully focused electron beam (beam diameter of approx. 70–80 nm) were used. The Count Rate was 1055.00 CPS. EDS studies of carbonized structures and sediment were performed with an FEI Inspect-S scanning electron microscope with an EDAX Apollo XV SDD EDS detector at 15, 10 and 5 keV acceleration voltage. Spectra were acquired for 30–90 s to obtain a good signal to noise ratio, and intensities were corrected with the ZAF algorithm. One piece of the specimen was sputter-coated with gold and scanned with high-voltage. EDS-SEM results can be taken from S1 Fig for carbonaceous material and from S2 Fig for calcitic fillings.

## Micro-Computed Tomography (Micro-CT)

Thirteen selected specimens were serially sectioned at the Core Facility for Micro-Computed Tomography (Vienna Micro-CT Lab), University of Vienna, Austria, using the custom-built VISCOM X8060 NDT (Germany) µ-CT scanner with different scan parameters, which delivers a stack of images with isometric voxel sizes. These are then combined, resulting in a 3D volume. Scan parameters for each specimen can be taken from S3 Table.

The specimens were segmented using the software Avizo Amira 2020 (Thermo Fisher Scientific) and virtually 3D-reconstructed in Drishti 2.7 [54].

## Mineralogical and geochemical studies

The composition of one specimen was analyzed by Raman spectroscopy at the Department of Mineralogy, University of Vienna. Raman spectroscopy was used for the in-depth characterization of the carbon phase of the fossil remains. The Raman spectra were obtained by means of a Horiba LabRAM HR Evolution system equipped with an Olympus BX-series optical microscope and Si-based charge-coupled, Peltier-cooled, device detector. Spectra were excited with 532 nm emission of a frequency-doubled Nd-YAG laser (calcite, 12 mW at the sample; graphitic carbon, 0.012 mW). A 50x lens (numerical aperture 0.55) was used to focus the light onto the surface of the sample. The light to be analyzed was dispersed with 1800 grooves per mm diffraction grating. Raman spectroscopy for the specimen NHMW 2021/0016/0397 (S3 Fig) revealed strongly disordered (i.e., sp$^2$ hybrid bonded) carbon as the material of the fossil structures. They were measured at low energies to avoid measuring the burning spot. The

fossils were measured in comparison to reference spectra of natural graphite from a uranium mineralization in Saskatchewan, Canada (for sample description see [55, 56]).

Elemental mapping with the Microprobe JEOL Hyperprobe JXA-8530F field emission electron microprobe (EMS) in combination with the online JEOL quantitative ZAF-correction program was conducted at the Central Research Laboratories of the NHMW on specimen NHMW 2012/0117/0024 (S1 Data).

### Measurements and statistics

Where possible, specimens were measured by using a digital caliper (S5 Fig, S4 Table). Further detailed measurements were carried out on original Micro CT data using the software Dragonfly Workstation Version 2021.1 and Avizo Amira V. 5.4.0. Measurements for comparison with recent *Sepia officinalis* were done on a dataset from Ziegler et al. [47]. Diameters of the fossil channel system were measured on original data of a specimen with a well-preserved canalicular system (NHMW 2012/0117/0028, at a resolution of 15 μm). Statistics were done with Microsoft Excel 2010.

### Thin-sectioning

Thin-sections from one specimen (NHMW 2012/0117/0024) were prepared at the laboratory of the Natural History Museum Vienna, Austria. The specimen was embedded in Araldite epoxy resin, then sectioned and mounted on slides for microscopy. The sections were polished with aluminium oxide and silicon carbide powders to a thickness of about 25 μm.

### Sections of recent squids

Sections of three dead squid *Loligo vulgaris* and two *Sepia officinalis* yielded deep insights into the morphology of cephalopod cartilage and enabled actuopaleontological comparisons. Anatomical sections of the coleoids were produced at the Department of Palaeontology, University of Vienna, Austria, and the cephalic cartilages were isolated (S4 Fig). Coleoids and cartilage were measured and photographed.

## Results

Within the Polzberg section, the investigated fossils were only found in the lower part of the section and thus seem to be limited to the more calcareous part (layers -50 to 0 cm, 60 to 80 cm, 80 to 100 cm, 100 to 120 cm and 140 to 160 cm) (log in Fig 1, S1 Table).

Two distinct, morphologically different types of black, amorphous, carbonized masses (Fig 3, S1 Fig, S1 Data) with a channel system (= canalicular system) were recorded–Type A (52 specimens) and Type B fossils (18 specimens). Sometimes, an associated "wing-like" structure occurs (18 specimens). Isolated "wing-like" elements only occur in three specimens. Fourteen fossils come from a private collection, 26 were new findings from recent excavations, 31 were excavated during field work in the 1990ies, 2 belong to historical material from the Natural History Museum Vienna (NHMW, Austria). The six examined specimens from the Geological Survey Austria (GBA) all come from Cave del Predil. 15 specimens (12 from Polzberg locality, three from Cave del Predil) featured associated, mainly in situ preserved belemnoid microhooks, eight specimens exhibited a phragmoteuthid proostracum. In collection material of the NHMW and the GBA we found only three slabs comprising all structures: belemnoid phragmocone and/or characteristic proostracum, belemnoid hooks and the here described carbonized fossils (GBA 2006/011/0020, NHMW 2005z0005/0021 and NHMW 2005z0005/0033).

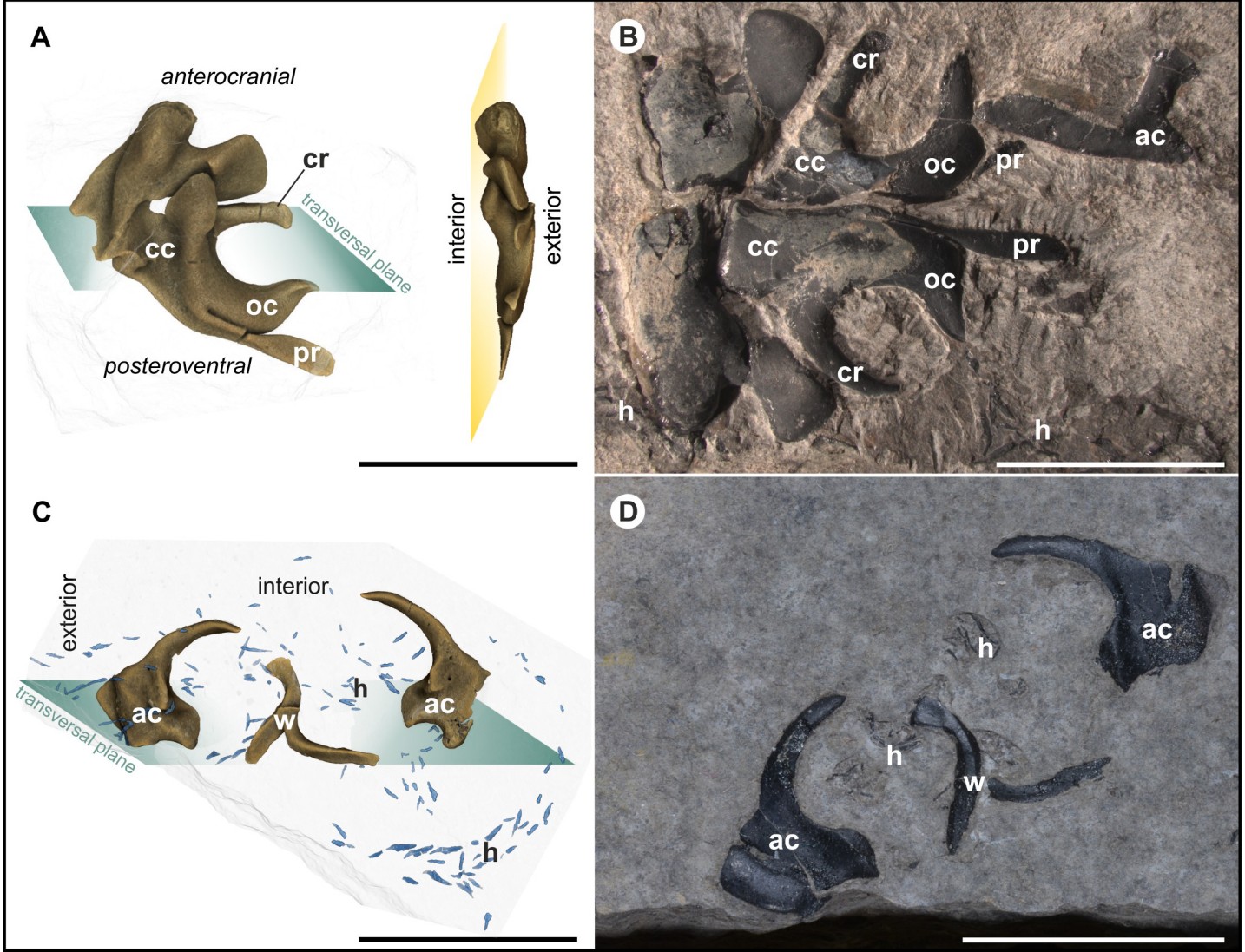

**Fig 3. Specimens of Type A and Type B fossils from the Polzberg *Konservat-Lagerstätte*.** (A) Suggested orientation for Type A fossils showing characteristic, curved C structure with prolonged processus. (B) Most complete specimen (NHMW 2021/0123/0057) showing symmetrical, semi-in situ configuration of the structures with likewise in situ preserved hooks, starting in the area of the long dorsal processus above the C structure. (C) Suggested orientation for Type B fossils showing a curved hook-like structure. (D) Cartilaginous structure associated with belemnoid hooks, NHMW 2021/0124/0003a, b. All scale bars 1 cm. ac arm cartilage, cc cephalic cartilage, cr carrier, oc ocular cartilage, h coleoid hooks, ac arm-cartilage structures, pr processus.

Three further specimens include carbonized fossil, phragmocone and proostracum, but lacking any hooks.

All specimens are surrounded by a calcitic layer (approx. 20 μm thick) and a preserved, calcite-filled channel system. Both types mainly appear in the lower, calcareous part of the section (S1 Table). S1 Table further summarizes the analytical methods used on the examined specimens from both localities. Measurements on the particular elements are listed in S4 Table.

## Type A fossil

Type A fossils (Figs 3A, 3B, 4A and 4B). Black, shiny, bilateral structure, maximum fossil lengths (diameter from most proximal point at processus to lowermost part of supposed

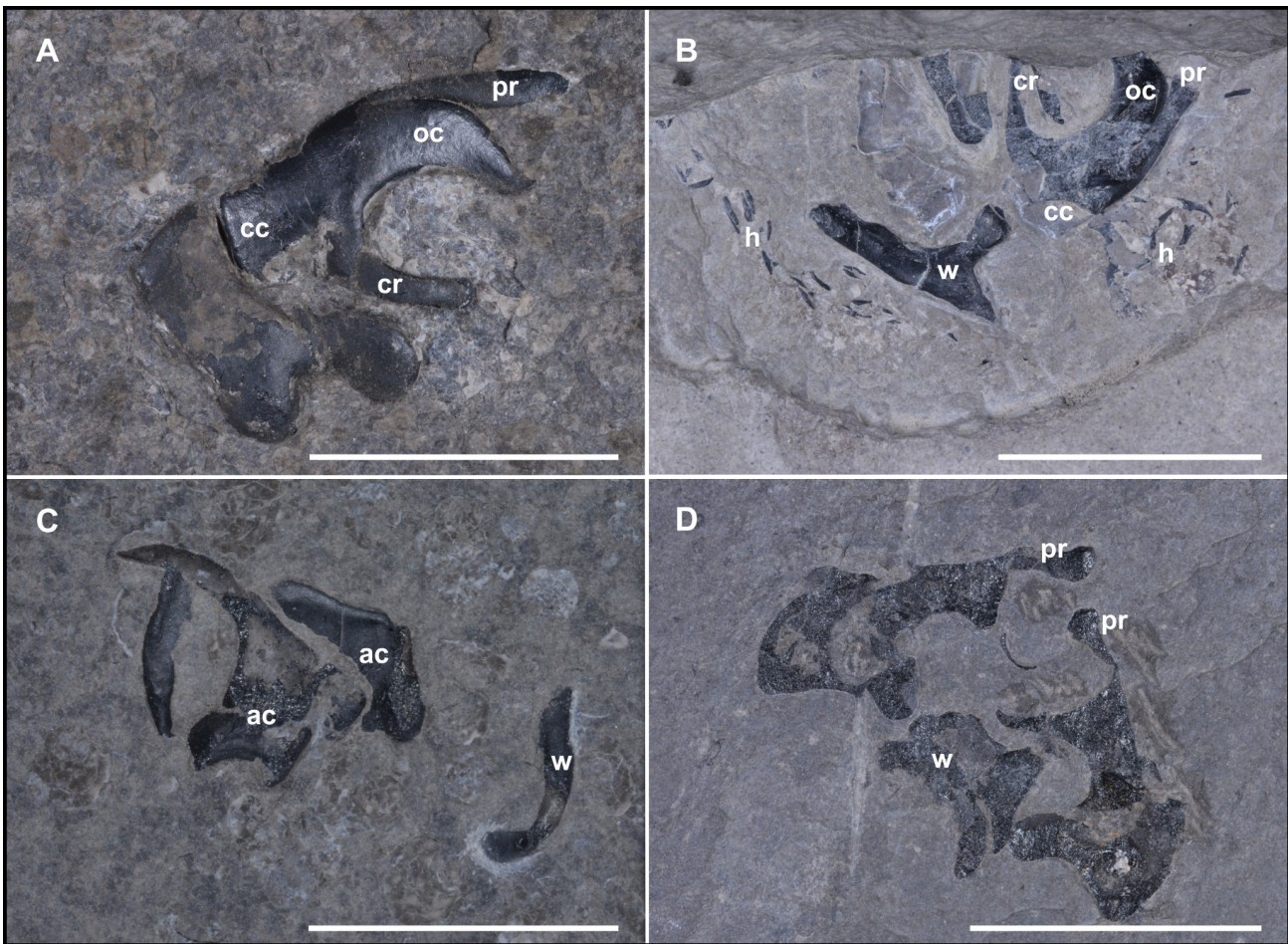

**Fig 4. Different views of Type A (possible cephalic cartilage) and B specimens. (A)** Standard "lateral" view of Type A fossils (NHMW 2012/0117/0001) with groove, characteristic processus on prominent C structure. **(B)** Supposed »cranial« view of Type A specimen (NHMW 2012/0117/0025) with associated wing-elements and in situ preserved hooks. **(C)** Two Type B fossils with a triangular base and two additional elongated elements (NHMW 2021/0124/0002). **(D)** Historical specimen from the Geological Survey Austria (GBA 2006/011/0041) from the Ringgraben ravine near Cave del Predil (Raibl, Italy). These specimens show a thicker and more prominent upper part of the C structure. cc cephalic cartilage, cr carrier, h belemnoid hooks, ac arm-cartilage, oc ocular cartilage, pr processus. All scale bars 1 cm.

"statocysts") range from 5 mm (single element) up to 24 mm (double elements); particular elements of the specimens such as "carrier"(Fig 3A and 3B) appear loosely connected; calculated volumes range from 31.82–253.95 mm$^3$ depending mainly on completeness of the available structure.

**Description.** The bilateral structures (Fig 3B) consist of black, shiny amorphous carbon. The surface bears several ridges and grooves (Fig 3A). A prominent C structure (average maximum height: 6.66 mm) with a thicker central part and thinner, proximal, curved ends is present in the center of the fossil. Opposite to the curved part, the opposing structure ("carrier", Fig 3A and 3B) comes relatively linear and only shows connections to other elements on lateral exterior. Some specimens reveal that the "carriers"are not necessarily fixed to the remaining structure, but more likely pass through it. The whole structure is penetrated by a widely distributed canal system whose canaliculi are calcite filled. The measured diameters of the calcite-filled (Fig 6C) channels vary from approx. 7 μm (peripheral) to approx. 100 μm in the central part. The most complete specimen (NHMW 2021/0123/0057; Fig 3B) has a maximum length

of 24.24 mm, with a height of 18.94 mm. 13 Type A fossils are associated with hooks, eight others with a belemnoid fan-shaped proostracum. Two specimens from Polzberg locality (NHMW 2012/0117/0025 and NHMW 2005z0005/0021) exhibit a cartilage with microhooks in association with an in situ preserved part of a proostracum. In this specimen there are also wing elements preserved from which the hooks start. Four specimens from Cave del Predil and three from Polzberg locality include all three features (phragmocone/proostracum, hooks and there here treated structures; see also S2 Table). When associated with a proostracum, this enigmatic C structure was found between 19.13 mm and 31.80 mm away from the last visible proostracum field (see also S2 Table). Generally, the distance between proostracum and C structure increases with increasing proostracum length. On average all black fossils are located 24.70 mm from the last visible field of the proostracum. Values for proostracum length to distance (black fossil and-proostracum) vary in a relatively narrow range from 1.18–1.54 (here only measurements of complete proostraca were taken into account).

Occasionally, "wing-like"structures (Fig 3C) are present (mainly associated with Type A fossils). As the C structures apparently grew proportionally, their maximum height (as figured in S5 Fig) was used as indicator for the full size. The present heights of the C structures vary within a range of 3.5 to nearly 10 mm (S4 Table), also depending on low-grade diagenetic distortion. Diagenetic cracks are visible. The processus (pr in Figs 3A and 4A) reaches lengths of 3.7–9.4 mm. The C structure-height to processus ratio ranges from 0.7–1.4. Associated wing-elements can be described as triangular-elongated in shape, with length-to-base ratios from 1.1–2.0.

## Type B fossil

Type B fossils (Figs 3C, 3D, 4C and 4D). Black, shiny structures, prolate shape with a slightly curved base (S5B Fig). Some specimens show an extended base. There are several ridges and grooves visible, channel system present. Large specimens reach maximum lengths of more than 10 mm (S4 Table). The distribution of the calcite filled channel system resembles that in Type A fossils but seems to be less developed. A prominent notch is present on the interior, but is absent on the exterior. These specimens are sometimes associated with in situ preserved belemnoid hooks and with one or two additional, elongated structures. Rough surface areas were mainly found on the posterior part of the fossil. The base widths of Type B fossils were measured from 3.3–7 mm, with heights from 6.6–10.5 mm. The volumes range from 26.13–32.76 mm$^3$ (S5 Table).

## Geochemistry and internal structure

Microprobe elemental mapping (Fig 6; S1 Data) of a thin-section at the fossil–sediment boundary indicates a high silicon (Si), aluminium (Al), potassium (K) and magnesium (Mg) content within the sediment. Distinct nodules of sulphur (S) and iron (Fe) are visible. Carbon (C) dominates the fossil structure, but carbon-seams are also present in the sediment. Interestingly, S, Al and titanium (Ti) limit another layer within the fossil structure. A transitional layer (approx. 5 μm), rich in Al forms the border to the outermost carbonized layer (see also Al image in S1 Data). The outermost carbonized layer (approx. 50 μm) has a slightly different geochemical composition than the inner carbonized structure. Noticeably the outermost carbon-layer of the fossil lacks sulphur (S), while the inner carbonized structure is rich in S, such as the sediment (S1 Data). Ti is generally just distributed the same as S. The carbonized fossil is surrounded by an external layer with high amounts of calcium (Ca) and oxygen (O). A canalicular system, also rich in Ca and O, draws through the fossil structure. Thin-sections (Fig 5), microprobe analysis (Fig 6; S1 Data) and serial sectioning confirm the prominent canal system

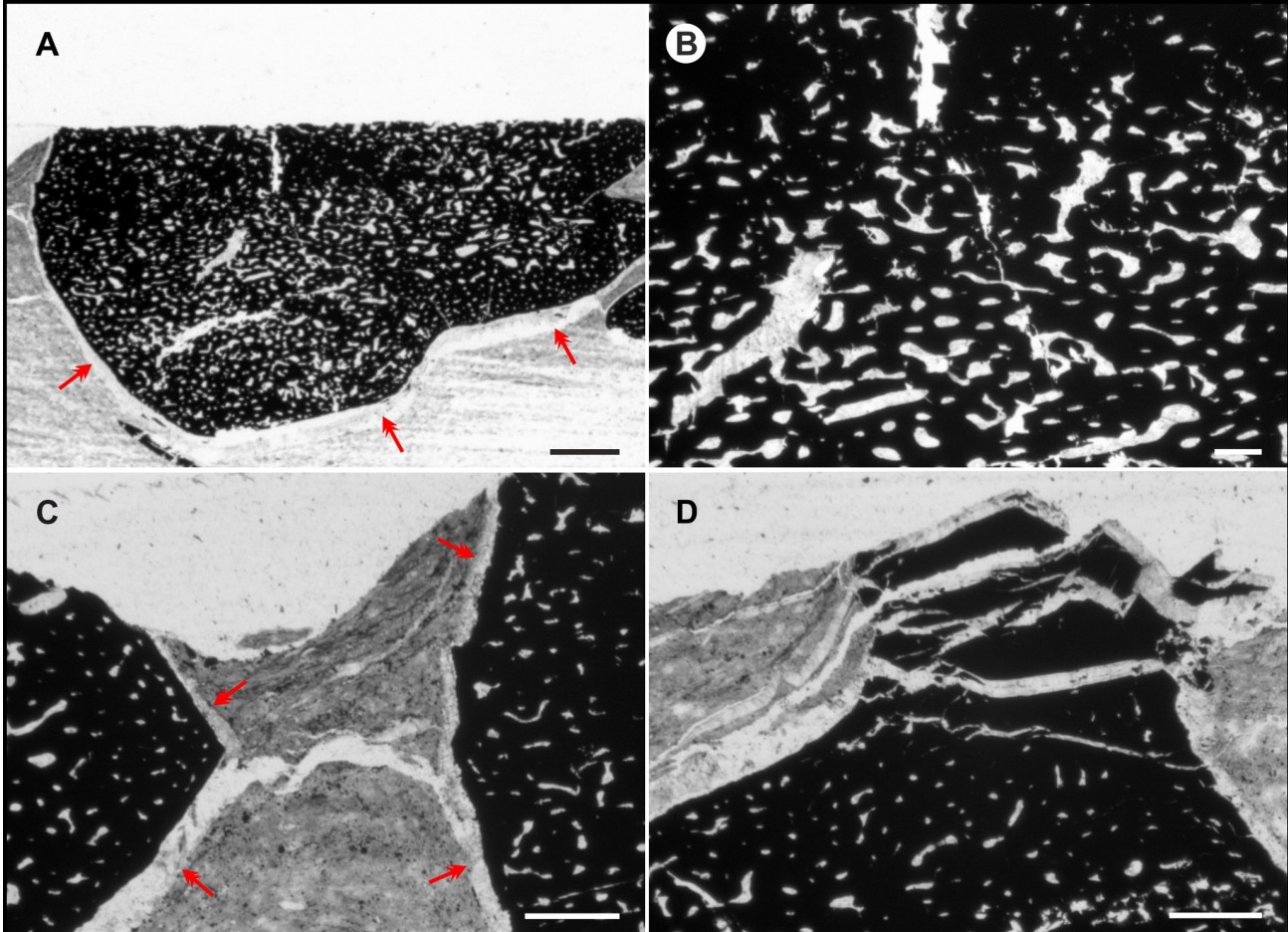

**Fig 5. Analyses carried out on specimen NHMW 2012/0117/0024. (A)** Thin-section of enigmatic black fossil reveals a widely distributed channel system (grey); red arrows: surrounding calcitic seam/layer. Note the curving of sediment around the fossil specimen. **(B)** Magnification showing pore space in the center of the fossil. **(C)** Red arrows mark the outer seam surrounding the fossil. **(D)** Diagenetic cracks in the black fossil structure. All scale bars 200 μm, except scale bar for A 500 μm.

that becomes denser toward the center of the structures. A diameter of up to 100 μm was measured for the central canals, 7–20 μm for proximal channels. The channel system follows the geometry of the fossils. The most distal areas of the fossils apparently lack distinct canals or are not preserved.

SEM images prove the presence of a distinct transitional layer between the carbonized fossil and calcareous sediment (Fig 7A). The thin layer surrounding all fossils varies from 20–100 μm (Fig 7B) and consists of the same material as the calcitic fillings (Fig 7C and 7D) of the channel system, while EDS-SEM analyses confirm that the fossils consist of carbon. Raman spectroscopy of one specimen (NHMW 2021/0016/0397) (S3 Fig) reveals a composition of strongly disordered (i.e., $sp^2$ hybrid bonded) carbon for the fossil specimens.

## Discussion

Due to the secondary carbonization, demineralization of the present structures in order to reinvestigate the original structure was unfeasible. Carbonized fossils are usually preserved as thin carbon-films. In contrary, the here described fossils are 3D-preserved, not flattened and show no traces of abrasion or other damage. They often occur with in situ preserved coleoid

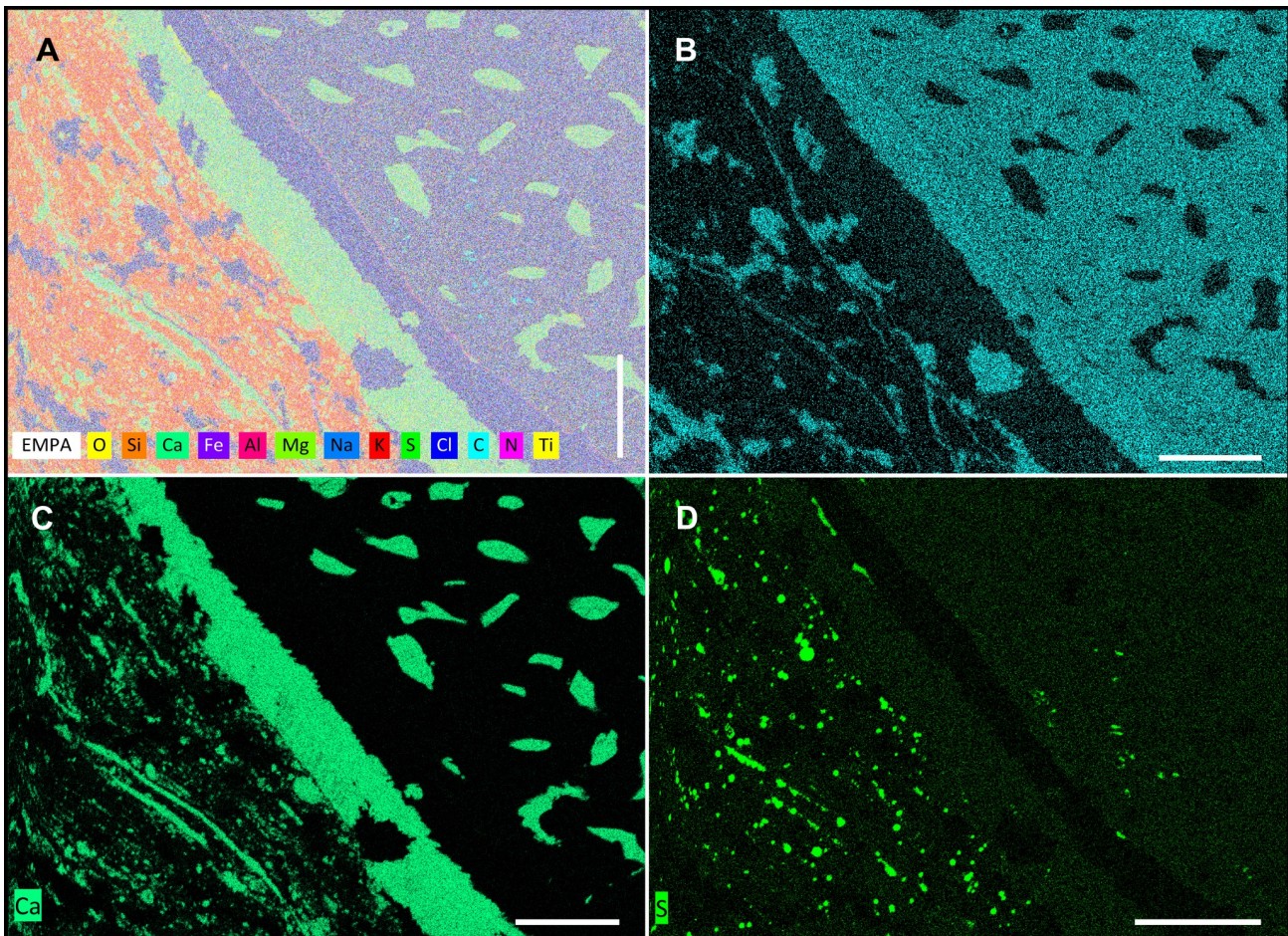

**Fig 6. Microprobe analysis reveals the elemental distribution near the sediment-fossil boundary. (A)** Composite image of elements in thin-section of specimen NHMW 2012/0117/0024. Aluminium, silicon and oxygen are the main sediment constituents. **(B)** Carbon is the dominant element in the black fossil structure. **(C)** Calcite-filled channels and the fossil surrounding calcitic seam/layer. **(D)** Distribution of sulfur in the sediment corresponds to pyrite nodules. Note the outer dark layer of the fossil, lacking sulfur. All scale bars 100 μm.

remains, such as phragmocone, three-parted proostracum and microhooks, from *Phragmoteuthis bisinuata*. The morphology of the particular elements and their internal structure can be homologized to modern coleoid cephalic cartilage. Mineralization is one possibility to preserve soft tissues such as cartilage in 3D. Vertebrates require Cholecalciferol (Vitamin D) uptake for their mineralization patterns. This substance is not found in squids [57]. Recent coleoids apparently do not need cartilage mineralization in their lifecycles. Nonetheless, in vitro, several experiments showed a principal ability for mineralization of coleoid cartilage under particular circumstances [52, 58, 59]. Experimental in vitro coleoid cartilage mineralization requires relatively high temperatures of 37˚C combined with a phosphate saturated environment [52]. Anyway, it cannot be excluded that the Carnian environmental conditions (including a low-oxygen environment with high amounts of sulphur) also favoured mineralization processes after the carcasses sank to the anoxic sea floor and decay began. A previous study proposed a bacterial pseudomorphosis of coleoid soft tissues from Polzberg [21]. We agree with that idea but suggest a prior mineralization process in particular of coleoid cartilaginous structures. The morphology of the fossils differs considerably from recent coleoid cephalic cartilage. Earlier studies indicated that the high contents of phosphatidylserine in

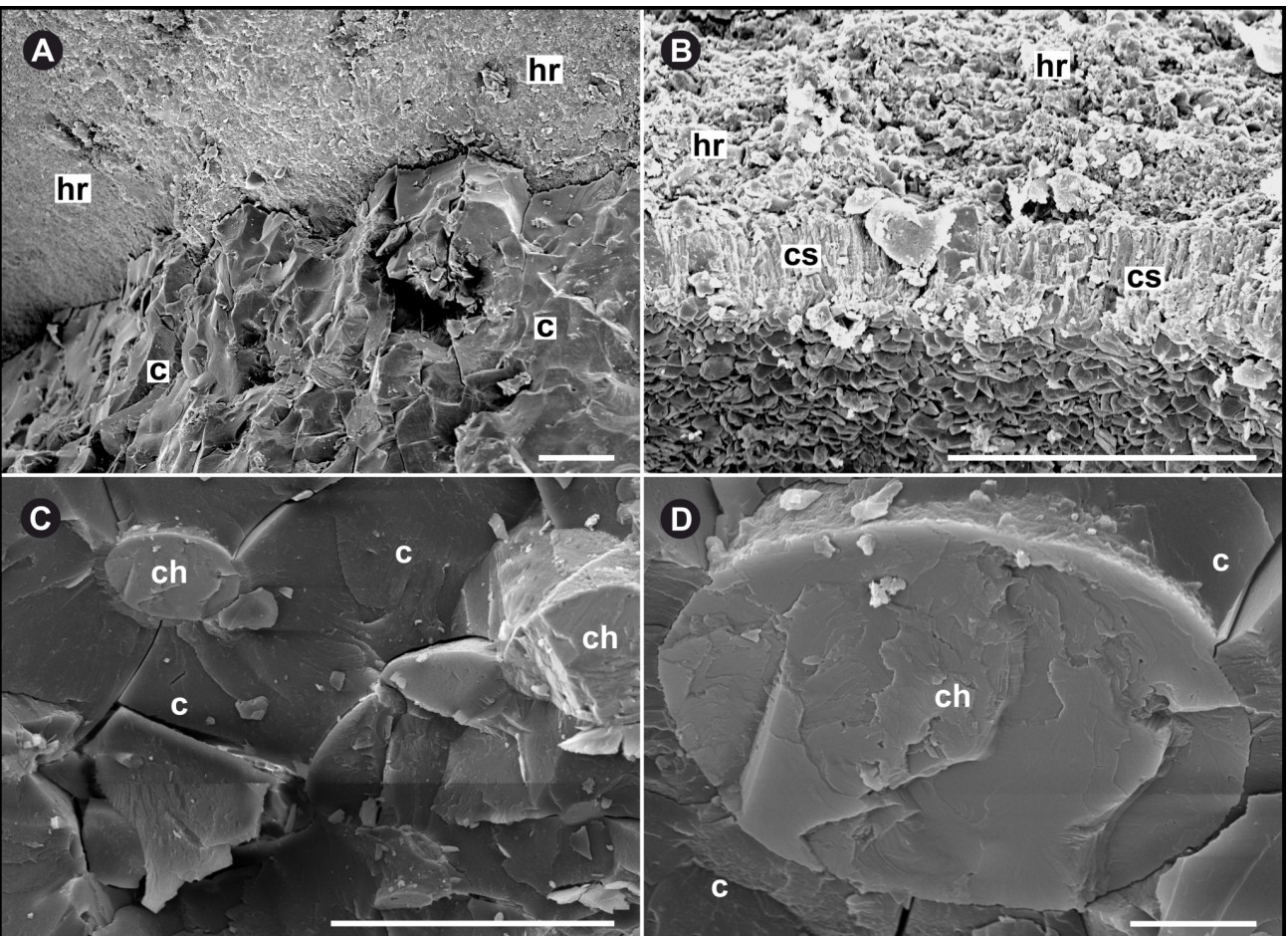

**Fig 7. EDS-SEM images of the ultrastructure of specimen NHMW 2012/0117/0024. (A)** Distinct boundary between fine calcareous host rock (hr) and carbonized, amorphous fossil (c). **(B)** Thin calcareous fossil-surrounding seam/layer (cs, approx. 20 µm thick). **(C)** EDX-SEM of amorphous structure with calcitic channel fillings. **(D)** Calcite-filled channel (magnification x 5000; scale bar 10 µm). hr host rock, c carbonized cartilage, ch calcitic channel fillings, cs calcareous fossil-surrounding seam. Scale bar except (D) 100 µm.

coleoid cartilage were associated with high mineralization rates [52]. As phosphatidylserine appears to be unevenly distributed within recent coleoid cartilage, this may result in incomplete mineralization rates and thus only partial preservation of cephalopod cartilage. The conspicuous, encapsulating outer calcitic layer, with abundant orthogonal cracks, is most probably caused by dehydration during carbonization and also hints at a prior mineralization of the embedded fossils. Local accumulations of pyrite confirms possible euxinic conditions.

## Type A fossils

The narrow range for the distance of the black fossil from the last field of the proostracum probably hints the affiliation of both structures. The black structures appear where a cephalic-cartilage-complex would have been expected. The size and morphology of the particular elements of Type A fossils closely resemble the cephalic cartilage-complex (including arm cartilage, ocular cartilage and cephalic cartilage with statocysts) of *Sepia officinalis* and other modern coleoid species (Fig 8). In particular, the characteristic shape of the C structure resembles slightly compressed ocular coleoid cartilage (oc in Fig 8). The ventral curvature of the C structure (Fig 8) is also identifiable in the ocular cartilage of *Loligo vulgaris*; even the prominent processus can be homologized to a

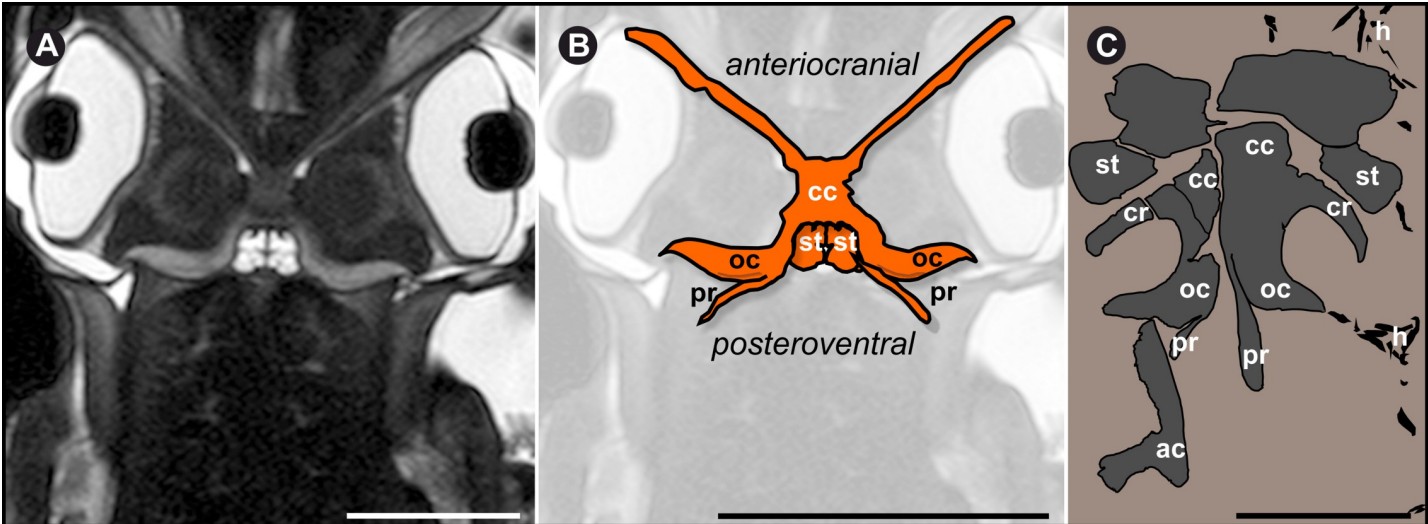

**Fig 8. Homologization of fossil specimens with modern coleoid cartilage.** **(A)** MRT-data [46] of *Sepia officinalis*. Note the hook-like ending and angle to the C structure. Scale bar 20 mm. **(B)** 2D-Drawing of cartilage, redrawn magnified from a, showing principal shape of the structure in *S. officinalis*. Scale bar 2 cm. **(C)** Digital drawing of most entire specimen (NHMW 2021/0123/0057) obtained during excavations in 2021, with suggested elements as labeled: ac arm cartilage, cc cephalic cartilage, cr carrier, h hooks, oc ocular cartilage, pr processus, st statocysts. Scale bar 1 cm.

corresponding structure in extant coleoids. The angle between the C structure and most probably very mobile processus is 114.4˚ in fossil specimens and 125.0˚ in recent cartilage. The preserved channel system shows a proximal–distal size-distribution, resembling the widely distributed channel system in the cephalic cartilage of recent cephalopods. More channels are present in the center of the cartilage, fewer more distally (Fig 5A). In fossils the channels with larger diameters are concentrated in the center of the structure.

Recent, dissected specimens of *L. vulgaris* showed mantle lengths of 142.3–152.5 mm and head cartilage sizes of about 20 mm (S5 Fig). Mantle lengths of dissected specimens of *S. officinalis* were 180–200 mm, with well-developed cephalic cartilage lengths around 25 mm. Overall, cartilage sizes varied strongly in size, shape and thickness. Moreover, the features of cephalic cartilage likely also vary during coleoid ontogeny and gender (own observations in dissected specimens of *L. vulgaris* and *S. officinalis*).

## Type B fossils

These symmetrical, introversively twisted specimens show conspicuous features such as the slight curvature of the prolongation to the interior (which is regarded as an essential difference to Type A fossils) and the prominent notch on the interior (Fig 3C and 3D). Overall, the cuvature of the structure differs strongly from Type A fossils. From a superficially viewpoint, these elements may resemble the megahook-like structures of brachial crowns which are known from some Mesozoic ammonoids [60, 61]. Moreover, some of these structures from Polzberg show an extended "base"(which more closely resembles cartilage-elements than hooks). The fossils from Polzberg locality show no basal openings as they are figured for some megahooks [19]. Due to the introversive twist, the present structures seem to be well suited for stabilizing tasks in distal body parts.

## Chemical composition

The elemental mapping (Fig 6; S1 Data) around the boundary between fossil and sediment hints the presence of increased amounts of clay minerals, which are known to be connected to

exceptional preservation in *Konservat-Lagerstätten* [62]. The carbonaceous composition of the fossil confirms the carbonization of the original structure. The high Ca and O in the enclosing layer around the fossil, as well as the canalicular system can be interpreted as secondary calcite, probably developed during shrinkage of the structure. Accumulations of Al are observed in systems of microbial fossilization, where Al-complexation probably prevented fast decay of macromolecular structures [63].

### Channel system

The channel system is significantly weaker developed in Type B fossils, than in Type A fossils. This points to a less needed supply and supports a more distal position in the phragmoteuthid body. A dense channel system in the centre of the fossil structure indicates that there is an increased need for supply in these regions, in contrast to distal regions. Both fossil types have a widely branched channel system that follows the geometry of the specimens. Three thick channels were observed in the center, one of them very prominent, accompanied by smaller peripheral channels. Channel diameters of about 150 μm have been reported from canalicular fin-cartilage of an early Triassic squid-like coleoid [38]. In recent *Loligo*, the extracellular matrix is penetrated by small canals with chondrocyte extensions [39]. Original sizes of the cartilaginous channel system are about one-hundredth of sizes given in [38]. Accordingly, the canals could have expanded due to the above-mentioned shrinking processes of the soft tissue during carbonization.

### Comparison with fossil specimens from Cave del Predil (Italy)

Specimens (GBA 2006/011/0003, GBA 2006/011/0012, GBA 2006/011/0020, GBA 2006/011/0028 GBA 2006/011/0041) (Fig 4D), all assigned to *Phragmoteuthis bisinuata* from late Triassic deposits of the Rinngraben ravine in Cave del Predil are known to resemble the Polzberg specimens in several ways. All the morphological elements from the Polzberg specimens could also be identified in objects from Rinngraben ravine in Cave del Predil. Overall, the Rinngraben ravine specimens show a poorer preservation and a more prominent appearance (see also the knob-like shape of the processus, Fig 4D). These slight morphological differences mainly reflect sedimentological and thus paleoenvironmental properties. Finally, the preservation of soft tissue in *Konservat-Lagerstätten* is influenced by many interconnected factors. This complicates any prognosis on how they will be preserved under particular circumstances.

### Homologization with the anatomy of modern coleoids

The anatomy of cephalic-ocular-cartilage-complexes in modern coleoid specimens such as of *S. officinalis* and *L. vulgaris* have been intensively studied. The perichondrium surrounds the whole cartilage elements and is more densely built than the remaining cartilage structure. Due to the symmetrical arrangement of the individual elements, homologization of the existing structures was simplified. Dimensions (see also S4 Table) and if available in situ positions of the fossil structures (S2 Table) are remarkably similar to corresponding extant coleoids. In coronal view, the cartilage-complex in Magnetic Resonance-Tomography (MRT) data from [46] reveals a C structure (Fig 7A and 7B), where the coleoid optical lobe rests in, similar to the conspicuous C structure of the fossils (Figs 3A and 8). The fossil Type A specimens (Fig 3A and 3B) show peripherally thickened regions on elements which are interpreted as the ocular cartilage (oc in Figs 3A, 3B, 4A and 4B), a feature also known in recent coleoids [53].

   Type B fossils (Fig 3C and 3D) are most likely disarticulated, distal parts of the arm cartilage (positioned at the anterocranial) which is, in extant coleoids, only loosely connected to the ocular-cephalic-cartilage-complex. The elements show prominent notches and grooves on the

internal side which can interact when assembled. The presence of comparable shapes (grooves, ridges, processes, "wing"-like structures; Figs 3 and 8) in ancient and recent specimens can probably be interpreted as muscular attachment sites. The well-developed canal system follows the shape of the fossils and indicates a cartilaginous origin. Channel sizes and their distribution within the fossil structures are comparable to modern cephalopod cartilage.

The characteristic bowl shaped eye cartilages protect the cephalopod optical lobes, see also Micro-CT data for *S. officinalis* from [47]. Just next to these lobes, the cephalopod brain rests in a cartilage structure, expanded to the statocysts (capsules with the aragonitic statoliths). The loosely connected arm cartilage appears as a stabilizing element for the coleoid arms. The elements of the cephalic-cartilage-complex have been described from numerous modern cephalopods [1–3, 5, 6, 47]. The fossil record documents the presence of corresponding structures in ancient cephalopod groups from the Paleozoic and the Mesozoic [8–14, 36, 37, 64–66].

Decay and decomposition of other anatomical parts lead to the disintegration and isolation of particular anatomical elements [19], which can probably result in slight differences of the finally preserved fossils. Variations in ontogenetic stages, along with geological deformation after burial, both contribute to the morphometric variations of these otherwise consistent structures. Although. Through time, there are differences in the morphological appearance of cephalopod cephalic cartilage it seems to be a constant element in cephalopod evolution, contributing to their predatory lifestyle. The lightweight brain-protecting structure supports high speed swimming abilities of modern coleoids. We see all criteria for homology fulfilled and interpret the present structures as mineralized and carbonized remains of belemnoid cephalic cartilage.

Although the specimens from Cave del Predil show slight morphological differences (which are most likely caused by diagenetic influences) to the ones from Polzberg locality, the same elements could be identified. Thus we also interpret them as mineralized and secondarily carbonized phragmoteuthid cephalic cartilage complex.

## Conclusions

Cephalopods have a long and rich evolutionary history with a high disparity of shapes and breakthrough evolutionary innovations. Even the morphologies of the cephalic cartilages of the various recent cephalopod groups differ from each other in several respects. We can therefore expect a multiplicity of that morphological disparity throughout the fossil record. The exceptional conditions during deposition resulted in excellent preservation in the Polzberg *Konservat-Lagerstätte*, enabling a detailed insight into the coleoid morphology. This is the first detailed report on the anatomy and ultrastructure of the 3D preserved cephalic cartilage from Polzberg locality. We analyzed the black amorphous structures with a multitude of methods including thin-sectioning, SEM imaging, microprobe analyses, RAMAN and EDS measurements. All of these methods point to the cartilaginous nature of these undetermined shapes. The carbonized elements exhibit multi-tube like structures filled by secondary calcite inside the main black material. Three semi-connected elements (C structure with processus, probable brain case and statocyst capsule; disarticulated arm cartilage; connective wing-element) form, doubled, an entire cartilage complex. Eleven analyzed specimens show symmetrical pairs of elements, with smooth surfaces and protrusions serving as muscle attachments, four of them also associated with wing-elements. Additionally, numerous elements are connected with or located close to arm hooks, and often crowded by belemnoid arm microhooks. Six specimens show the entire belemnoid body with phragmocone and/or proostracum, cartilage and arm hooks, indicating the nature and subsequently the exact position of the black carbonized elements. The described cartilage complex is direct evidence for invertebrate cartilage in

Mesozoic belemnoids and hence of phragmoteuthids in the Triassic ocean. The development of cartilaginous structures in marine invertebrates is still a matter of intensive discussion. The distribution of cartilage is meaningful for taxonomic questions, whereby evidence for cartilage in fossil cephalopods can help to clarify the evolutionary role of cartilage in invertebrate groups.

We present a possible solution for a 150-year-old paleontological discussion on the taxonomical affiliation of these carbonized fossils and suggest that the material represents the preservation of a mineralized and secondarily carbonized cephalic-ocular-arm-cartilage complex of the belemnoid *Phragmoteuthis bisinuata*. Due to the secondarily carbonized preservation style of the present fossils (no demineralization possible), this interpretation is mainly based on morphological studies in fossil and extant cephalopod groups.

In a next step, the segmentation and visualization of the canalicular system will be necessary for a conclusive explanation of the interconnected relationship between coleoid biology, cartilage mineralization processes, environmental conditions and diagenesis.

## Supporting information

**S1 Fig. SEM-EDS report for sample NHMW 2012/0117/0024 (Carbon).**
(PDF)

**S2 Fig. SEM-EDS report for sample NHMW 2012/0117/0024 (calcitic fillings).**
(PDF)

**S3 Fig. Raman spectrum for specimen NHMW2021/0016/0397 measured at low-energy conditions.**
(PDF)

**S4 Fig. Cephalic cartilage of *Loligo vulgaris*. A.** Specimen of *Loligo vulgaris*, red arrows marking position of cephalic cartilage. **B.** Well developed cephalic cartilage of *Loligo vulgaris* specimen from cranial view, yellow arrow pointing to opening for oesophagus. C Cephalic cartilage from lateral view, exhibiting ocular cartilage. cc cephalic cartilage; oc ocular cartilage. Scale bars 1 cm.
(PDF)

**S5 Fig. Visualization of metrics, measured on specimens.** $h_c$ height of C structure; $l_p$ length of processus; $m_h$ height of megahook; $m_b$ base-length of megahook. Scale bars 1 cm.
(PDF)

**S1 Table. List of examined samples.** The features of examined specimens and conducted methods, as well as associated belemnoid remains (phragmocone, proostracum, hooks) are given. The term indet. fossil in the table refers to here described black structures. Fifty-nine samples stem from Polzberg locality, seven from Rinngraben ravine near Cave del Predil (Julian Alps, Italy). Indicated are inventory numbers, locality, Type A or Type B fossil and/or wing, the applied methods (Micro-CT scanning resolution is given in brackets) and preserved belemnoid features. In recently collected specimens, the outcrop layer is given. NHMW corresponds to all inventory numbers except where GBA is given. Multiple elements on one slab are not separately mentioned;; PO Polzberg main section, ms measurements.
(PDF)

**S2 Table. Metrics of black fossil structures in association with coleoid remains.** $foss_l$ full length of visible phragmocone-proostracum; $p_w$ width of phragmocone; $pro_l$ length proostracum; d distance last field of proostracum–fossil structure remarks. All measurements in mm.

Only specimens listed, where measurements were possible.
(PDF)

**S3 Table. Micro-CT Scan parameters for fossil specimens.**
(PDF)

**S4 Table. Measurements conducted on fossil specimens.** Each specimen of each sample is listed here measured separately; $h_c$ height of C structure; $l_p$ length of processus; $l_w$ length of wing; $w_b$ base-length of wing element; $ac_h$ height of arm cartilage; $ac_b$ base-length of arm cartilage; all measurements in mm. Elements could not be measured in GBA specimens and are therefore not listed. Only specimens listed, where measurements were possible.
(PDF)

**S5 Table. Volumes of fossil specimens, obtained from Micro-CT data.**
(PDF)

**S1 Data. Microprobe report for sample NHMW 2012/0117/0024.**
(PDF)

## Acknowledgments

We are grateful to Franziska and Hermann Hofreiter (Gaming), the owner of the Polzberg section for the digging permission during the whole duration of the project. We thank Birgitt and Karl Aschauer (Waidhofen an der Ybbs) for providing many fossil specimens for detailed scientific investigations. We especially thank Dan Topa (SEM, microprobe), Anton Englert (thin-sections), Goran Batic (mineralogical thin-sections) for their technical support and Lutz Nasdala (all Vienna) for carrying out Raman Spectroscopy. Valentin Blüml and Christina Kaurin (both Vienna) for segmentation of Micro-CT data. Leon Ploszczanski and Matthias Kranner (both Vienna) for support with SEM pictures. Martin Zuschin (Vienna) is thanked for dedicated supervision. We especially thank the editor Steffen Kiel (Stockholm), Christian Klug (Zürich) and an anonymous reviewer for constructive and helpful comments on the manuscript.

## Author Contributions

**Conceptualization:** Alexander Lukeneder.

**Data curation:** Petra Lukeneder, Alexander Lukeneder.

**Formal analysis:** Petra Lukeneder.

**Funding acquisition:** Petra Lukeneder, Alexander Lukeneder.

**Investigation:** Petra Lukeneder.

**Methodology:** Petra Lukeneder.

**Project administration:** Alexander Lukeneder.

**Resources:** Alexander Lukeneder.

**Visualization:** Petra Lukeneder.

**Writing – original draft:** Petra Lukeneder.

**Writing – review & editing:** Petra Lukeneder, Alexander Lukeneder.

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
