## [Decision Letter · Decision Letter 0]

4 Mar 2022

PONE-D-22-03905Mineralized coleoid cranial cartilage from the Late Triassic Polzberg *Konservat-Lagerstätte* (Austria)*PLOS ONE*

*Dear Dr. Lukeneder,*

*Thank you for submitting your manuscript to PLOS ONE. After careful consideration, we feel that it has merit but does not fully meet PLOS ONE’s publication criteria as it currently stands. Therefore, we invite you to submit a revised version of the manuscript that addresses the points raised during the review process.* *See "Additional Editor Comments" below.*

*Please submit your revised manuscript by Apr 18 2022 11:59PM. If you will need more time than this to complete your revisions, please reply to this message or contact the journal office at plosone@plos.org. When you're ready to submit your revision, log on to https://www.editorialmanager.com/pone/ and select the 'Submissions Needing Revision' folder to locate your manuscript file*.

*Please include the following items when submitting your revised manuscript:**A rebuttal letter that responds to each point raised by the academic editor and reviewer(s). You should upload this letter as a separate file labeled 'Response to Reviewers'.**A marked-up copy of your manuscript that highlights changes made to the original version. You should upload this as a separate file labeled 'Revised Manuscript with Track Changes'.**An unmarked version of your revised paper without tracked changes. You should upload this as a separate file labeled 'Manuscript'.***If applicable, we recommend that you deposit your laboratory protocols in protocols.io to enhance the reproducibility of your results. Protocols.io assigns your protocol its own identifier (DOI) so that it can be cited independently in the future. For instructions see: https://journals.plos.org/plosone/s/submission-guidelines#loc-laboratory-protocols. Additionally, PLOS ONE offers an option for publishing peer-reviewed Lab Protocol articles, which describe protocols hosted on protocols.io. Read more information on sharing protocols at https://plos.org/protocols?utm_medium=editorial-email&utm_source=authorletters&utm_campaign=protocols*.

*We look forward to receiving your revised manuscript*.

*Kind regards*,

*Steffen Kiel, Ph.D*.

*Academic Editor*

*PLOS ONE*

*Journal Requirements:*

2. *Please review your reference list to ensure that it is complete and correct. If you have cited papers that have been retracted, please include the rationale for doing so in the manuscript text, or remove these references and replace them with relevant current references. Any changes to the reference list should be mentioned in the rebuttal letter that accompanies your revised manuscript. If you need to cite a retracted article, indicate the article’s retracted status in the References list and also include a citation and full reference for the retraction notice.*

3. In your manuscript, please provide additional information regarding the specimens used in your study. Ensure that you have reported specimen numbers and complete reposiPONE-D-22-03905tory information, including museum name and geographic location. 

For more information on PLOS ONE's requirements for paleontology and archaeology research, see https://journals.plos.org/plosone/s/submission-guidelines#loc-paleontology-and-archaeology-research.

5. We note that Figure 1 and SI Fig 1 in your submission contain [map/satellite] images which may be copyrighted. All PLOS content is published under the Creative Commons Attribution License (CC BY 4.0), which means that the manuscript, images, and Supporting Information files will be freely available online, and any third party is permitted to access, download, copy, distribute, and use these materials in any way, even commercially, with proper attribution. For these reasons, we cannot publish previously copyrighted maps or satellite images created using proprietary data, such as Google software (Google Maps, Street View, and Earth). For more information, see our copyright guidelines: http://journals.plos.org/plosone/s/licenses-and-copyright.

a. You may seek permission from the original copyright holder of Figure 1 and SI Fig 1 to publish the content specifically under the CC BY 4.0 license.  

*Additional Editor Comments:*

*Hallo Petra,*

*beide Gutachter waren von dem Manuskript begeistert, haben aber auch reichlich Anmerkungen und Verbesserungsvorschläge gemacht, die es nun einzuarbeiten gilt. Ich freue mich auf eure revidierte Version!*

*Viele Grüsse,*

*Steffen*

**

*Reviewers' comments:*

*Reviewer's Responses to Questions*

*

**Comments to the Author**
*

*1. Is the manuscript technically sound, and do the data support the conclusions?*

*The manuscript must describe a technically sound piece of scientific research with data that supports the conclusions. Experiments must have been conducted rigorously, with appropriate controls, replication, and sample sizes. The conclusions must be drawn appropriately based on the data presented. *

*Reviewer #1: Yes*

*Reviewer #2: Partly*

*2. Has the statistical analysis been performed appropriately and rigorously? *

*Reviewer #1: N/A*

*Reviewer #2: N/A*

*3. Have the authors made all data underlying the findings in their manuscript fully available?*

*The PLOS Data policy requires authors to make all data underlying the findings described in their manuscript fully available without restriction, with rare exception (please refer to the Data Availability Statement in the manuscript PDF file). The data should be provided as part of the manuscript or its supporting information, or deposited to a public repository. For example, in addition to summary statistics, the data points behind means, medians and variance measures should be available. If there are restrictions on publicly sharing data—e.g. participant privacy or use of data from a third party—those must be specified.*

*Reviewer #1: Yes*

*Reviewer #2: Yes*

*4. Is the manuscript presented in an intelligible fashion and written in standard English?*

*PLOS ONE does not copyedit accepted manuscripts, so the language in submitted articles must be clear, correct, and unambiguous. Any typographical or grammatical errors should be corrected at revision, so please note any specific errors here.*

*Reviewer #1: Yes*

*Reviewer #2: Yes*

*5. Review Comments to the Author*

*Please use the space provided to explain your answers to the questions above. You may also include additional comments for the author, including concerns about dual publication, research ethics, or publication ethics. (Please upload your review as an attachment if it exceeds 20,000 characters)*

*Reviewer #1: Dear authors,*

*congratulations to this nice piece of research!*

*It contains a comprehensive suite of methods and I agree with the interpretations and conclusions!*

*There are only a few aspects I am not happy with, most importantly, I think that the main finding is the interpretation that these structures are indeed cephalic cartilages. Thus, I recommend the following actions:*

*1. Expand the chapter where you homologize structures (the conclusion is almost longer than your discussion of this aspect).*

*2. Really discuss homology: relative position, specific quality and evolutionary transitions. It is not that I am not believing you, your discussion just lacks some detail.*

*3. Do this for the main parts of the cartilage.*

*4. Add one phrase where you mention this since I think this is one of the main discoveries! I was always wondering what these black blobs were, now we know!*

*Besides, I marked some style issues. For example, I would not use 'clearly', since this is rhetoric. Also, Fig. 8 is strange, because panel B expands beyond its frame. I marked other small issues in the pdf.*

*Overall, this shouldn't be too much work.*

*I look forward to see this published!*

*Best wishes,*

*Christian (Klug)*

*Reviewer #2: Dear authors,*

*congratulations for this very informative contribution about phragmoteuthid coleoids from the Triassic of Austria. Fossil cranial cartilages of the last common ancestor of octopuses, squids and cuttlefish in 3d preservation is extraordinary.*

*I find your manuscript well written, your methods are appropriate, your descriptions concise, and your figures of sufficient quality.*

*I fully agree with your interpretation whereupon your “type A” represents cranial cartilage, but I am not conform with your idea of mega-hooks in the case of “type B” (see my detailed comments annotated in the enclosed pdf). Please note, that both types exhibit similarities & may reflect two different cranial morphologies. If you persist, the presence of mega-hooks in phragmoteuthids allows an array of new palaeontological implications that still need to be addressed.*

*Apart from this, I think some parts of the manuscript are improvable in terms of a better reading (e.g., sometimes your terminology is inconsistent; more see my detailed comments annotated in the enclosed pdf).*

*After minor revisions the ms should be considered for publication.*

*with best wishes*

*Dirk*

*6. PLOS authors have the option to publish the peer review history of their article (what does this mean?). If published, this will include your full peer review and any attached files.*

**

**

*Reviewer #1: **Yes: **Christian Klug*

*Reviewer #2: No*

**

*While revising your submission, please upload your figure files to the Preflight Analysis and Conversion Engine (PACE) digital diagnostic tool, https://pacev2.apexcovantage.com/. PACE helps ensure that figures meet PLOS requirements. To use PACE, you must first register as a user. Registration is free. Then, login and navigate to the UPLOAD tab, where you will find detailed instructions on how to use the tool. If you encounter any issues or have any questions when using PACE, please email PLOS at figures@plos.org. Please note that Supporting Information files do not need this step.*

---

## [Author Response · Author response to Decision Letter 0]

11 Mar 2022

Dear Editor, dear Reviewers,

thank you for the valuable comments on our manuscript. It has greatly benefited from your suggestions and we hope our changes are appropriate. We included our detailed responses to the decision letter to the "Response to Reviewers" file. We have extended the homologization section and due to your input came back to our original idea that Type B fossils are disarticulated parts of the arm cartilage. We have updated our references and corrected mistakes. 

We hope that you are satisfied with our adaptations and hope that you will come to a positive decision for a publication in PLOS ONE.

Yours sincerely,

Petra Lukeneder & Alexander Lukeneder

---

## [Editor Report · Decision Letter 1]

15 Mar 2022

PONE-D-22-03905R1Mineralized belemnoid cephalic cartilage from the Late Triassic Polzberg * Konservat-Lagerstätte* (Austria)PLOS ONE

Dear Dr. Lukeneder,

Thank you for submitting your manuscript to PLOS ONE. After careful consideration, we feel that it has merit but does not fully meet PLOS ONE’s publication criteria as it currently stands. Therefore, we invite you to submit a revised version of the manuscript that addresses the points raised during the review process.

See details below.

We look forward to receiving your revised manuscript.

Kind regards,

Steffen Kiel, Ph.D.

Academic Editor

PLOS ONE 

Journal Requirements:

Additional Editor Comments:

Dear Petra,

I am now happy with the science but there are still a number of editorial issues. I have uploaded an annotated manuscript, please check it carefully. The main issues are:

- take care of the numerous typos;

- make sure you stick to either American or British spelling, but don't mix them;

- in some of the figures, some letters indicating the panel numbers are really difficult to read; please add a 'background box' or some other means to improve their readability.

- in the reference section, article titles are not capitalized but spelled in the correct way, even if all words start with a capital letter in the original title;

- also in the reference section, please check carefully if the generic names are in italics; I highlighted some, but other may have escaped my attention.

Once these issues are taken care of, I'll be happy to accept your interesting manuscript!

Cheers, Steffen
---

## [Author Response · Author response to Decision Letter 1]

15 Mar 2022

Dear Steffen, 

thank you very much for your helpful comments on our manuscript! We have done all the corrections, especially all spelling mistakes. We have substituted the rest of the British expressions (except in institutional names of course) and corrected our reference list. The only suggestion which we have not adapted so far is line 687, because “Bromalites” is not a genus name, but a collective designation.

Affiliation of the corresponding author was adapted.

Best regards,

Petra Lukeneder & Alexander Lukeneder

---

## [Editor Report · Decision Letter 2]

17 Mar 2022

Mineralized belemnoid cephalic cartilage from the Late Triassic Polzberg *Konservat-Lagerstätte* (Austria)

PONE-D-22-03905R2

Dear Dr. Lukeneder,

We’re pleased to inform you that your manuscript has been judged scientifically suitable for publication and will be formally accepted for publication once it meets all outstanding technical requirements.

Kind regards,

Steffen Kiel, Ph.D.

Academic Editor

PLOS ONE
---

## [Editor Report · Acceptance letter]

29 Mar 2022

PONE-D-22-03905R2 

Mineralized belemnoid cephalic cartilage from the Late Triassic Polzberg *Konservat-Lagerstätte* (Austria) 

Dear Dr. Lukeneder:

I'm pleased to inform you that your manuscript has been deemed suitable for publication in PLOS ONE. Congratulations! Your manuscript is now with our production department. 

Kind regards, 

on behalf of

Dr. Steffen Kiel 

Academic Editor

PLOS ONE